



# DMS cycling in the Sea Surface Microlayer in the South West Pacific: 1. Enrichment potential determined using a novel sampler

Alexia D. Saint-Macary[1,2], Andrew Marriner[1], Theresa Barthelmeß[3], Stacy Deppeler[1], Karl Safi[4], Rafael Costa Santana[1,2], Mike Harvey[1] and Cliff S. Law[1,2]

[1]National Institute of Water and Atmospheric Research, Wellington, 6021, New Zealand
[2]Department of Marine Science, University of Otago, Dunedin, 9016, New Zealand
[3]GEOMAR Helmholtz Centre for Ocean Research Kiel, Kiel, 24148, Germany
[4]National Institute of Water and Atmospheric Research, Hamilton, 3216, New Zealand

*Correspondence to*: Alexia D. Saint-Macary (alexia.stmac@gmail.com) and Cliff S. Law (cliff.law@niwa.co.nz)

**Abstract.** Elevated dimethyl sulfide (DMS) concentrations in the sea surface microlayer (SML) have been previously related to DMS air-sea flux anomalies in the South West Pacific. To further address this, DMS, its precursor dimethylsulfoniopropionate (DMSP), and ancillary variables were sampled in the SML and also subsurface water at 0.5 m depth (SSW) in different water masses east of New Zealand. Despite high phytoplankton biomass at certain stations significant chlorophyll *a* and DMSP enrichments were only apparent at one of six stations, with the DMSP enrichment factor (EF) ranging from 0.81 to 1.25. DMS in the SML was determined using a novel gas-permeable tube technique which measured consistently higher concentrations than with the traditional glass plate technique; however, DMS enrichment was also present at only one station, with the EF ranging from 0.40 to 1.22. SML DMSP and DMS were influenced by phytoplankton community composition, with correlations with dinoflagellate and *Gymnodinium* biomass*,* respectively. DMSP and DMS concentrations were also correlated between the SML and SSW, with the difference in ratio attributable to greater DMS loss to the atmosphere from the SML. DMS in the SML did not significantly influence regional DMS emissions, with the calculated air-sea DMS flux of 1.0 to 11.0 µmol m$^{-2}$ d$^{-1}$ consistent with climatological estimates for the region. These results extend previous regional observations that DMS is associated with dinoflagellate abundance but indicate that additional factors are required for significant enrichment in the SML.

## 1 Introduction

Dimethyl sulfide (DMS), a trace gas derived from dimethylsulfoniopropionate (DMSP) produced by phytoplankton (Keller et al., 1989), is a natural aerosol precursor (Yu and Luo, 2010; Sanchez et al., 2018), and a potential regulator of climate. DMS is ventilated to the atmosphere and oxidized to non-sea salt sulfate aerosols and methane sulfonic acid, which subsequently contribute to formation and growth of cloud condensation nuclei (CCN). Condensation of water vapor on CCN leads to the formation of cloud droplets, with the resulting increase in cloud reflectivity potentially reducing incoming solar radiation to the ocean and consequently phytoplankton growth and DMS emissions, as postulated by the CLAW hypothesis (Charlson et



al., 1987). Although the CLAW hypothesis has been questioned, due to spatial and temporal decoupling of CCN and DMS emissions, and the identification of other CCN precursors (Quinn and Bates, 2011), it continues to be investigated to elucidate feedbacks.

DMS concentrations in the surface ocean fluctuate in response to variation in regional biology and physical controls (Stefels et al., 2007). DMSP concentration is influenced by phytoplankton community composition (Keller et al., 1989), bacterial processes (Curson et al., 2017), grazing (Wolfe et al., 1994), and physicochemical variables such as nutrient availability, light, salinity and temperature via DMSP and DMS cycling (Stefels et al., 2007). These factors may have a direct effect on DMSP production and consumption, and also an indirect effect via their influence on plankton community composition (Stefels et al.,

2007; Stefels, 2000). Variability in DMSP and DMS in the surface ocean is reflected in regional variation in DMS flux to the atmosphere. Generally, the air-sea flux is estimated from DMS concentration in surface waters (2 to 10 m), but there is evidence that processes within the sea surface microlayer (SML) may also affect the flux (Walker et al., 2016). The SML is vertically less than 1,000 µm and connects the ocean to the atmosphere (Hunter, 1980). Biogeochemical cycling within the SML may differ to that of the subsurface water (SSW) due to the concentration of biogenic material and exposure to high irradiance, both

of which influence dissolved trace gas concentrations and flux to the atmosphere (Upstill-Goddard et al., 2003; Carpenter and Nightingale, 2015), and production of primary and secondary aerosols (Leck and Bigg, 2005; Roslan et al., 2010). DMS enrichment in the SML relative to the SSW has been reported, with an enrichment factor (EF) range of 0.6 to 5.7 (Yang et al., 2005a; Zhang et al., 2009; Walker et al., 2016; Yang, 1999). DMS enrichment is often associated with blooms of certain phytoplankton groups, such as dinoflagellates and haptophytes (Yang, 1999; Matrai et al., 2008; Yang et al., 2009; Walker et

al., 2016), whereas enrichment is often absent where diatoms dominate (Zhang et al., 2008; Matrai et al., 2008), except when present in high abundance (Yang et al., 2005a; Zhang et al., 2009). High DMS enrichment in the SML has also been reported in association with specific physical and meteorological conditions, and may result in anomalously high air-sea DMS flux and discrepancies between observed and calculated DMS air-sea fluxes (Marandino et al., 2008; Walker et al., 2016).

A global DMS climatology model based on all reported observations (82,996 datapoints; (Wang et al., 2020)), shows a seasonal pattern, particularly in mid to high latitude regions (Kettle et al., 1999). The climatological average DMS concentration in the South West Pacific does not exceed 4 nmol $L^{-1}$, except during January and February when DMS concentration ranged between 6 and 10 nmol $L^{-1}$. East of New Zealand, the Subtropical (STW) and Subantarctic (SAW) water masses meet at the Subtropical front (STF) along the Chatham Rise, where high phytoplankton production is often observed (Murphy et al., 2001; Chiswell

et al., 2015). The STW north of the Chatham Rise is characterized by warm saline water and low phytoplankton productivity due to low nitrogen availability, whereas the SAW south of the Chatham Rise is fresher with high macronutrient concentrations but low productivity due to iron limitation (Boyd et al., 1999). Consequently, this region provides an ideal area to determine the influence of variability in water mass properties on DMS and aerosol precursor production (Law et al., 2017). During the SOAP (Surface Ocean Aerosol Production) voyage in the Chatham Rise region in 2012, DMSP and DMS distribution varied



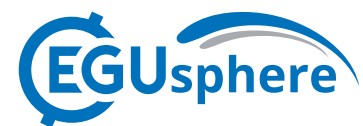

with phytoplankton composition and biomass, with elevated DMS concentrations relative to regional climatological estimates (Bell et al., 2015; Walker et al., 2016; Wang et al., 2020). DMS concentrations exceeded 20 nmol L$^{-1}$, resulting in an elevated DMS flux during a dinoflagellate bloom (Bell et al., 2015; Walker et al., 2016; Lizotte et al., 2017; Lawson et al., 2020), with two independent approaches indicating that DMS enrichment in the SML influenced air-sea flux (Walker et al., 2016).

However, the SOAP results also raised questions regarding the maintenance of DMS enrichment in the SML and its influence on DMS emissions. Sampling of the SML is challenging and existing techniques are not optimal for trace gas sampling. The Garret screen (Garrett, 1965) has generally been preferred to the plate (Harvey and Burzell, 1972) for DMS sampling of the SML (Yang et al., 2001), although this may result in artefacts (Yang et al., 2005b; Walker et al., 2016), and underestimation of DMS concentration (Yang and Tsunogai, 2005; Zhang et al., 2008; Zemmelink et al., 2006; Matrai et al., 2008). To address

this, a novel SML sampling technique using gas-permeable tube to minimize DMS loss was deployed, and results compared to those obtained with the glass plate method during the Sea2Cloud voyage. The primary aim of this voyage was to examine the relationships between marine biota and aerosol formation (Sellegri et al., in revision), and so DMSP, DMS and ancillary variables were measured in the SML and SSW to generate EFs, and establish the factors influencing DMS cycling and emission (see companion paper, Saint-Macary et al). Estimation of DMS fluxes enabled reconciliation of regional estimates based upon

empirical data (Bell et al., 2015; Walker et al., 2016) and climatology models (Lana et al., 2011; Wang et al., 2020).

## 2    Method

### 2.1    Regional setting

The Sea2Cloud voyage took place on 16 to 28 March 2020 (austral autumn) onboard R/V *Tangaroa* in the Chatham Rise region (Figure 1a). The characteristics of the water masses sampled during this voyage and meteorological conditions are

summarized in Table 1, and detailed in the Sea2Cloud introduction paper (Sellegri et al., in revision). Six workboat deployments were carried out to sample the SML and SSW in different water mass types: STF at stations 1 and 2, SAW at stations 3 and 4, STW at station 5 (see Figure 1a, Table 1). Mixed water (Mixed) at station 6 was a composite of coastal and shelf water from Cook Strait and STW, with higher nutrient content than STW, as presented in Figure 1b. SSW measurements were complemented by surface water collection using a CTD. Local wind measurements were obtained from an Automatic

Weather Station (AWS) located at 25.2 m above sea level above the bridge of the R/V *Tangaroa*, which was exposed to all wind directions (Smith et al., 2018).



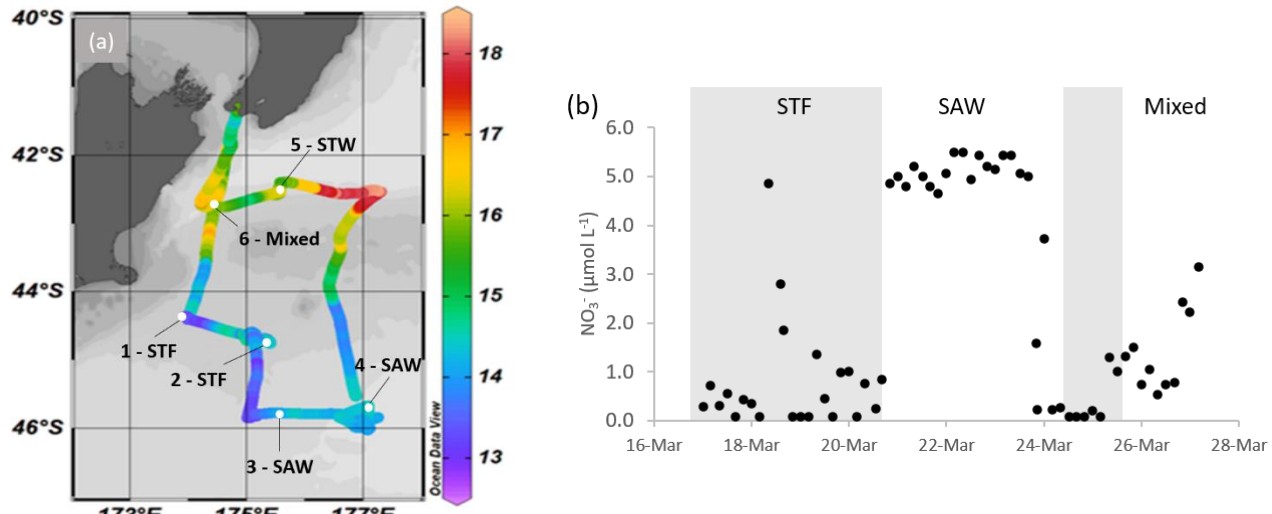

**Figure 1: (a) Sea2Cloud voyage track overlain by sea surface temperature (°C), using Ocean Data View (Schlitzer and Reiner, 2020), with workboat station positions identified. The grey shading shows the undersea topography, with the darker grey band along 43.5°S indicating the Chatham Rise. (b) Nitrate concentration during the Sea2Cloud voyage, measured at 5 m.**

**Table 1: Summary of environmental conditions during the workboat deployments. Water side variables were determined from the vessel underway system which sampled at 5 m depth, windspeed was measured by AWS, and presented as the average ($\pm$ sd) over the previous 12 h before sampling.**

| Date | Latitude (South) | Longitude (East) | Workboat station and water masses | Sampling time $t_0$-$t_{end}$ (NZDT) | Average wind speed (m s$^{-1}$) | Temp. (°C) | Salinity | Chl-$a$ (µg L$^{-1}$) | Dominant phytoplankton group (carbon) at 5 m depth |
|---|---|---|---|---|---|---|---|---|---|
| 18 March | 44°24'331 | 173°58'134 | 1-STF | 0900-1050 | 3.8 ($\pm$ 2.2) | 13.03 | 34.55 | 1.54 | Diatoms |
| 19 March | 44°45'234 | 175°24'173 | 2-STF | 0830-1034 | 7.5 ($\pm$ 0.9) | 14.15 | 34.44 | 3.64 | Diatoms |
| 21 March | 45°48'590 | 175°08'826 | 3-SAW | 1020-1159 | 7.9 ($\pm$ 2.5) | 13.37 | 34.33 | 0.37 | Dinoflagellate |
| 23 March | 46°00'053 | 177°04'637 | 4-SAW | 0845-1022 | 7.4 ($\pm$ 2.6) | 13.94 | 34.36 | 0.43 | Dinoflagellate |
| 25 March | 42°34'940 | 175°29'901 | 5-STW | 1533-1714 | 5.4 ($\pm$ 2.8) | 16.18 | 34.88 | 1.02 | Diatoms |
| 26 March | 42°45'043 | 174°20'006 | 6-Mixed | 0950-1138 | 8.2 ($\pm$ 3.6) | 16.24 | 34.78 | 0.89 | Diatoms |

## 2.2 Sampling of the SML

The SML and SSW were sampled from a workboat at distance from the R/V *Tangaroa* between 0800 and 1100 (all times
NZDT) on each day except for 5-STW (see Table 1), during periods when the wind speed was below 10 m s$^{-1}$. DMS was sampled using a novel gas-permeable tube technique in which a 280 cm long loop of silicone tube (external diameter 2.41 mm, wall thickness 0.49 mm) was deployed on the sea surface. The gas-permeable tube was filled with Milli-Q® water (MQ) prior to deployment and closed by joining the two tube ends with a union. The gas-permeable tube was threaded through floating



beads to ensure contact with the SML and deployed free-floating upstream of the workboat. Once in contact with the SML,
the technique relies upon diffusion of DMS through the gas-permeable tube membrane across the concentration gradient
between seawater and MQ. In theory at least 50% of the tube surface area is in contact with the SML and surface seawater,
with the remainder exposed to the atmosphere. The gas-permeable tube was recovered after 10 minutes, with the MQ
withdrawn immediately using a syringe and stored in a chilly bin. SML sampling was carried out in duplicate at each station.

Prior to deployment in the field, the diffusion efficiency of the gas-permeable tube was determined in semi-controlled
conditions using coastal seawater in Wellington, New Zealand, naturally elevated in DMS (range: $1.25 - 16.88$ nmol L$^{-1}$,
average 4.94 nmol L$^{-1}$). The calibration tank was continuously filled with seawater at a flowrate of 75 L min$^{-1}$, with a constant
overflow to ensure that there was no SML formation; this approach resulted in a uniform and homogenous DMS concentration
in the tank for the gas-permeable tube floating to equilibrate with. The gas-permeable tube was filled with MQ and placed on
the surface of the seawater in the tank for 10 minutes, after which the MQ was withdrawn into a syringe with no headspace
whilst the gas-permeable tube remained in contact with the surface water. The 10-minute exposure time was pre-determined
in laboratory experiments and represented the optimum time to achieve significant diffusion efficiency whilst reducing
deployment time. The gas-permeable tube was then removed from the water and refilled with MQ with the experiment repeated
3 to 8 times. In addition, ambient seawater in the calibration tank was sampled at $t_0$ and $t_{+10min}$ for each repetition. Between
each repetition, samples were transferred to the laboratory for immediate analysis. The DMS diffusion efficiency was
subsequently determined using:

$$D = \frac{[DMS]_{MQ}}{[DMS]_{tank}} \times 100 \qquad (1)$$

where $[DMS]_{MQ}$ and $[DMS]_{tank}$ are the DMS concentrations in nmol L$^{-1}$ measured in the MQ and the calibration tank,
respectively. The average D for 10 minutes exposure was 61% ($\pm$ 10% S.D, n=19) as determined over a 4-month period during
which the seawater temperature range was similar to that during the Sea2Cloud voyage, at $12 - 16$ °C. Further details of the
gas-permeable tube technique are provided in (Saint-Macary, 2022). The average D was then applied to calculate the actual
DMS concentration in the SML, $[DMS]_{SML}$, using:

$$[DMS]_{SML} = [DMS]_{MQ} \times \frac{100}{D} \qquad (2)$$

where $[DMS]_{MQ}$ is the DMS concentration in the MQ after 10 minutes of exposure in the SML.

A glass plate (Harvey and Burzell, 1972) and a sipper were also used for sampling of DMSP, DMS and ancillary variables in
the SML. The sipper consists of a tube with multiple inlets that float on the sea surface. A syringe was used to slowly draw
SML water through the open inlets to sample for chlorophyll *a* (chl-*a*), phytoplankton composition and DMSP. The sipper
external diameter was similar to the gas-permeable tube (2.2 and 2.4 mm, respectively), so enabling sampling of a similar SML
thickness but larger SML water volume in a shorter period. Samples for SML bacterial abundance were collected using a glass



plate, as described in Zäncker et al. (2017), and DMSP and DMS were also sampled with the plate for method comparison. The reproducibility of the gas-permeable tube, plate and sipper were calculated using the duplicate measurements for DMSP and DMS, following:

$$\text{reproducibility} = \frac{|(\text{duplicate a} - \text{duplicate b})|}{\text{average of a and b}} \times 100 \tag{3}$$

For DMSP and DMS sampling of the SSW, a Teflon tube was deployed with the inlet at a depth of approximately 0.5 m by a system of ropes and fishing weights. Fifty millilitres of SSW were withdrawn using a syringe and collected in an amber bottle leaving no headspace. For larger volumes for other ancillary variables in the SSW, a bottle was immersed to 0.5 m below the surface and filled with seawater. To avoid SML contamination, the bottle was immersed with its lid on, then opened and closed in the SSW before recovery. For each variable the enrichment factor (EF) was calculated by dividing the concentration in the 145 SML by its concentration in the SSW.

### 2.3 CTD sampling

The CTD was launched after the workboat sampling around 1130 at each station. Six depths from 5 to 150 m were sampled with 12 L Niskin bottles, although only the results from 5-m depth are discussed in this paper. For DMS sampling from the CTD casts, the water was overflowed by gravity by at least 100% into amber bottles and then sealed with no headspace.

### 2.4 DMSP and DMS analytical system

For DMS measurements, water from the amber bottles was withdrawn in plastic Terumo® syringes. The samples were injected through a 25-mm glass microfiber filter (GF/F) into a 1-mL loop, before transfer to a silanized sparging tower, where the sample was sparged for 5 minutes with nitrogen ($N_2$) at a flow rate of 50 mL min$^{-1}$. Nafion® dryers removed the water vapor from the gas samples before DMS preconcentration at $-110\,°C$ on a Tenax® trap. The trap was then heated to $120\,°C$ to 155 release the DMS onto an Agilent Technology 6850 Gas Chromatography coupled to an Agilent 355 Sulfur Chemiluminescent Detector (GC-SCD). The daily sensitivity and detection limit of the detector were confirmed using VICI® methyl ethyl sulfide and DMS permeation tubes. The average detection limit during the voyage was 0.14 ($\pm$ 0.03) pgS sec$^{-1}$. For total DMSP measurements, 20 mL glass vials were filled with seawater and 2 pellets of NaOH added before gas-tight sealing the vials, which were stored at ambient temperature in the dark. DMSP was analysed one day after sampling using the same semi-160 automated purge and trap system followed by GC-SCD, as described above. A wet standard calibration curve was made daily from a stock solution of DMSP diluted in MQ, with calibration concentrations ranging from 0.1 to 95 nmol L$^{-1}$. These were decanted into 20 mL gas tight glass vials, hydrolysed with 2 pellets of NaOH and then injected into the sparging unit and processed as with the samples.



## 2.5 Ancillary variables

For chl-*a* analysis, 250 mL of seawater was filtered onto a 25-mm GF/F filter, and then stored at −80 °C until analysis. Chl-*a* was extracted in 90% acetone, measured and compared with chl-*a* standards by spectrofluorometry using a Varian Cary Eclipse fluorometer, with an accuracy of 0.5 nm at 541.2 nm. An acidification step was used to correct for pheophytin interference (10200 PLANKTON).

Phytoplankton community structure was determined for cells >5 µm using a Flowcam (Fluid Imaging Technologies Inc). Two hundred and fifty millilitres of seawater was filter concentrated using a 47-mm diameter 3-µm polycarbonate filter to 10 mL final volume and stored at 4 °C until analysis. One millilitre of 25× concentrated seawater sample was run through a 80-µm depth Field of View flow cell (FC80FV) at 0.050 mL min$^{-1}$ and 20 frames per second, with an imaging efficiency of $61.9 \pm 2\%$. Images were taken using a 10× objective on AutoImage mode. Total run time for each sample was 20 min. Between 4-SAW

and 5-STW, the sample volume and flow rate were increased to 2 mL at 0.100 mL min$^{-1}$, with an imaging efficiency of 32.7%, due to the high abundance of large diatoms (e.g. *Chaetoceros* sp.). Images were classified into cell size and class groupings using VisualSpreadsheet v4.16.7 software, by size category (<10 µm; 10 to 20 µm; 20 to 50 µm and >50 µm), and the results given as total phytoplankton biovolume of each size class.

For microscopic analysis of phytoplankton community composition, 500 mL of seawater was preserved at 1% (final concentration) Lugol's iodine solution, with samples stored at room temperature in the dark. Phytoplankton community composition and cell numbers for phytoplankton >5 µm were determined using optical microscopy, following the method described in Safi et al. (2007) and references herein. Briefly, 100 mL subsamples were settled for 24 hours and the supernatant then carefully syphoned with 10 mL transferred to Utermohl chambers and resettled (Edler and Elbrächter, 2010). Where

possible, all abundant organisms were identified to genus or species level before being counting. Phytoplankton biovolume estimates were calculated from the dimensions of each taxa and approximated geometric shapes (spheres, cones, ellipsoids) initially following Olenina (2006). The biovolumes were subsequently used to calculate cell carbon (mg C m$^{-3}$) using equations from the literature; Olenina (2006) and Montagnes and Franklin (2001) for diatoms, and Olenina (2006) and Menden-Deuer and Lessard (2000) for dinoflagellates and nanoflagellates. Menden-Deuer and Lessard (2000) was also applied to other low

biomass unidentified groups referred to as small flagellates.

## 2.6 DMS air-sea flux calculation

The DMS air-sea flux, F, was calculated using the gas transfer flux equation, following Eq. (4):

$$F = k_{DMS,COARE} \times \left( [DMS]_w - \frac{[DMS]_{atm}}{H} \right) \tag{4}$$



with H the Henry's law solubility coefficient for DMS (Dacey et al., 1984), $[DMS]_w$ dissolved DMS concentration, $[DMS]_{atm}$ DMS concentration in the atmosphere, and $k_{DMS,COARE}$ the gas transfer coefficient. The latter was calculated using the NOAA COARE gas transfer (COAREG) version 3.6 algorithm (Fairall et al., 2003; Fairall et al., 2011) and parameterized in terms of local wind speed scaled to 10 m height, as described in Bell et al. (2015). The gas transfer velocity was adapted for DMS using the Schmidt number (Sc) calculated using local temperature (T) in °C (Saltzman et al., 1993), following Eq. (5).

$$Sc = 2674.0 - 147.12 \times T + 3.726 \times T^2 - 0.038 \times T^3 \tag{5}$$

The atmospheric DMS concentration $[DMS]_{atm}$ was neglected as this is several orders of magnitude lower than the dissolved DMS concentration (Kremser et al., 2021). Flux estimates were obtained using DMS concentrations from three different depths: $F_{SML}$ corresponds to DMS air-sea flux calculated using DMS concentration in the SML, $F_{SSW}$ to DMS concentration in the SSW, and $F_{5\,m}$ to DMS concentration at 5 m depth from the CTD.

## 2.7    Statistical analysis

The Shapiro test was used to verify the normality of variable distribution. For the non-normally distributed variables Spearman's rank correlation was carried out and for the normally distributed data a Pearson test was applied. Linear correlation was considered significant where the coefficient of correlation (rho and r for Spearman's rank and Pearson tests, respectively) was higher than 0.5 and p-value was lower than 0.05.

## 3    Results

### 3.1    Comparison of plate, sipper and gas-permeable tube

The reproducibility of SML techniques is generally not reported, although this is critical, particularly as the width and presence of the SML is inherently patchy and heterogenous (Frew et al., 2002; Ribas-Ribas et al., 2017). The lowest reproducibility for DMSP for the six workboat stations was obtained using the plate, with a median of 7% (interquartile range: 4 – 8% n = 6), relative to the sipper which had a median reproducibility of 3% (interquartile range: 1 – 7% n = 5) (Figure 2a). The reproducibility determined for DMSP using the sipper was subsequently applied in the current study to the other ancillary variables chl-*a*, DMSP and phytoplankton that were sampled by the same method to identify a significance threshold. This resulted in significant difference between the SML and SSW only where EF values were outside 0.97 – 1.03 (corresponding to sipper reproducibility of ± 3%). For DMS, the reproducibility of the plate and gas-permeable tube were similar although the plate had a smaller interquartile range (plate median 7%; interquartile range: 5 – 9% n = 6; gas-permeable tube median 8%; interquartile range: 3 – 18% n = 6; Figure 2b), and so a similar approach was applied for the DMS significance threshold, with no significant difference between SML and SSW DMS when EF was within 0.92 – 1.08.



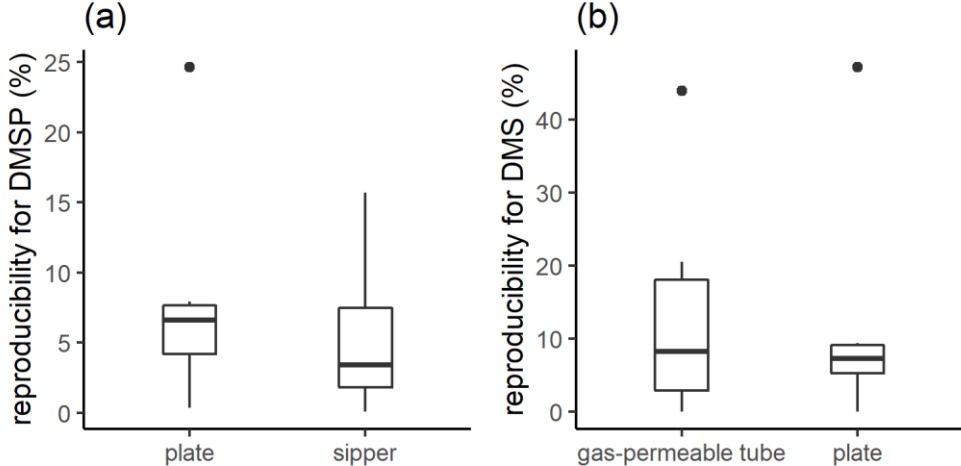

**Figure 2: Box plot of the reproducibility for (a) DMSP measurements with the plate and the sipper, and (b) DMS measurements with the gas-permeable tube and the plate. The boxplot represents the distribution of the data, with the box corresponding to the interquartile range and the bold horizontal line the median. The limits of the vertical lines represent the upper and lower fences. The outliers are represented by points outside the fences.**

### 3.2 DMSP and DMS in the SML and SSW

DMSP concentration was highest in the SML and SSW of STF with an average of 76 nmol L$^{-1}$ (Figure 3a) and lowest at STW and Mixed water at 32 and 20 nmol L$^{-1}$, respectively. The average EF DMSP was 0.93 (range: 0.81 – 1.25) with enrichment only observed at 5-STW (Figure 3b). Sampling with the plate showed a similar spatial trend to the sipper, but with lower average EF DMSP of 0.67 (range: 0.55 – 0.91), and no enrichment of DMSP at any station. The higher DMSP concentrations with the sipper may reflect that this method samples some water from immediately below the SML, whereas the plate only withdraws the organic layer associated with the SML (Harvey and Burzell, 1972; Cunliffe and Wurl, 2014).



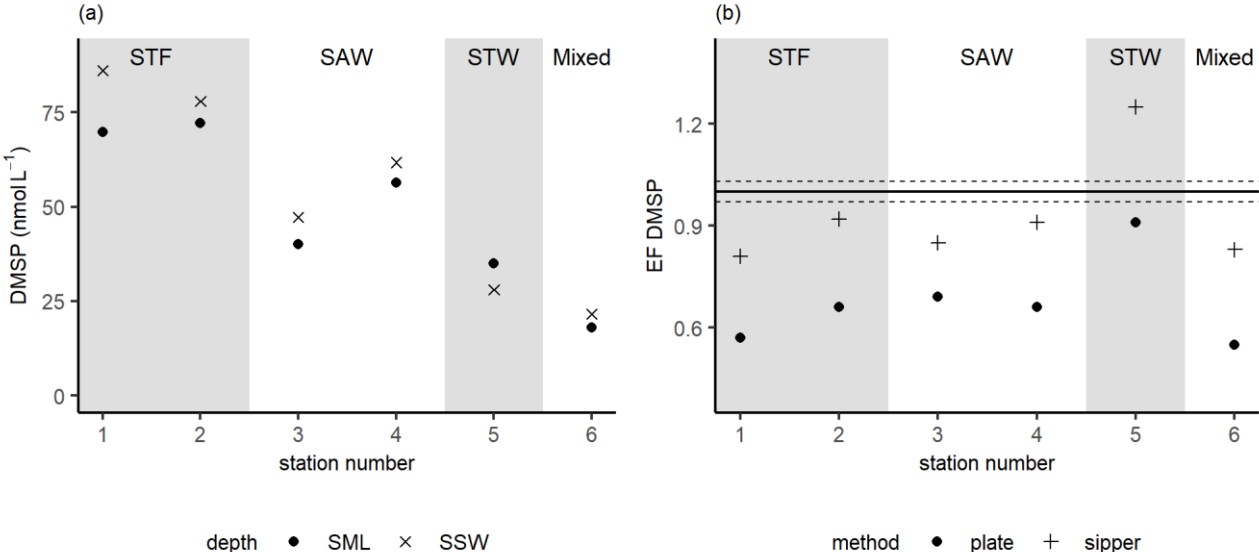

**Figure 3: (a) DMSP concentrations, sampled in the SML by the sipper, and in the SSW, and (b) EF DMSP from the sipper and plate. The dashed lines represent the EF significance threshold determined from the reproducibility of the sipper (3%). Water mass type is indicated by the labels at the top of the figure and also the shading.**

Three stations, 3-SAW, 5-STW and 6-Mixed had relatively low DMS concentrations of ~1.5 nmol L$^{-1}$ with no significant difference in concentration between the SML, SSW and 5 m depth (Figure 4a). In contrast, DMS concentration in the SSW was generally higher at the other 3 stations, ranging from 4.2 to 6.4 nmol L$^{-1}$ whilst concentrations in the SML and 5 m were similar, indicating SSW maximum in DMS. The gas-permeable tube showed no DMS enrichment at 5 of the 6 stations, with only 3-SAW showing significant SML enrichment. The overall average EF DMS was 0.83 (range: 0.40 – 1.22), with 3 stations showing DMS depletion in the SML. Conversely, when the plate was used to sample the SML significant depletion in DMS was apparent at all stations, with an average EF DMS of 0.46 (range: 0.28 – 0.68; Figure 4b), suggesting loss of DMS by sampling with the plate.



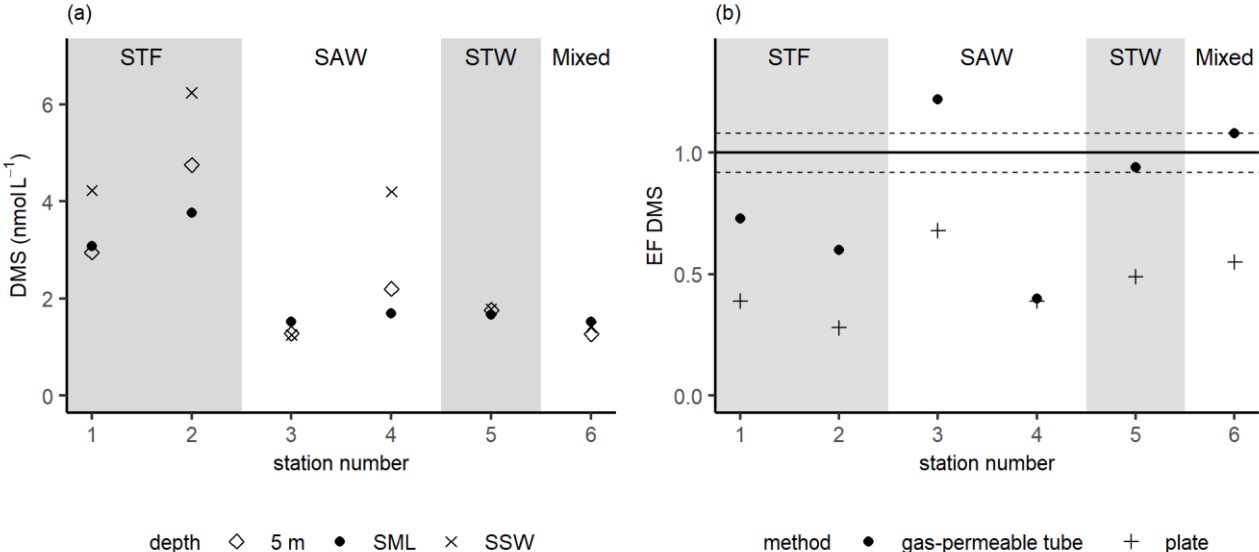

**Figure 4: (a) DMS concentrations in the SML, SSW, and 5 m, and (b) EF DMS determined by the gas-permeable tube and glass plate. The dashed lines represent the significance threshold determined from the median reproducibility of the gas-permeable tube (8%). Water mass type is indicated by the label at the top of the figure and also the shading.**

## 3.3    Ancillary variables

### 3.3.1    Chl-*a*

Highest chl-*a* concentrations (~4.3 µg L$^{-1}$) were found at 2-STF in the SML and SSW, with lower uniform chl-*a* concentrations (average 0.5 µg L$^{-1}$) at all three surface depths at 3-SAW, 4-SAW and 6-Mixed (Figure 5a). Average EF chl-*a* was 1.06 (range: 0.50 – 2.90), with significant enrichment in the SML only at 3-SAW (EF = 1.06), whereas the EF at the remaining stations was within the error for the sipper (Figure 5b). However, average EF chl-*a* in the SML relative to 5 m depth was 1.22, (range: 0.68 – 2.88) with very high enrichment at 2-STF, indicative of a near-surface chl-*a* gradient.



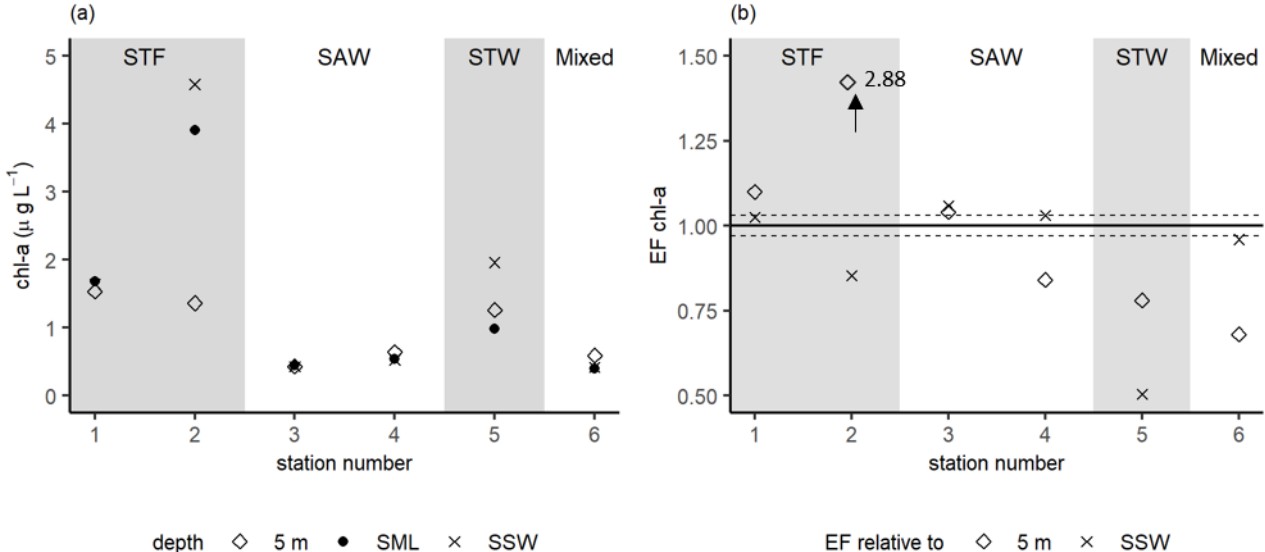

**Figure 5: (a) Chl-*a* concentrations in the SML, SSW, and 5 m depth. (b) Chl-*a* enrichment in the SML relative to the SSW and 5 m, with the dashed lines representing the significance threshold determined from the reproducibility of the sipper (3%). Chl-*a*** 260 **enrichment relative to 5 m at 2-STF is off-scale at 2.88 as indicated by the vertical arrow. Water mass type is indicated by the label at the top of the figure and also the shading.**

### 3.3.2 Phytoplankton community

Phytoplankton abundance, as determined using the Flowcam, is described in terms of total biovolume (>5 µm) and also for the separate size fractions (<10 µm, 10-20 µm, 20-50 µm and >50 µm) (Figure 6). Total phytoplankton biovolume was highest

265 at 2-STF ($8.55\times10^8$ to $1.13\times10^9$ µm³ L⁻¹), and lowest at 3-SAW and 6-Mixed ($2.93\times10^7$ to $8.16\times10^7$ µm³ L⁻¹). Station 1-STF displayed high biovolume in the SML but low biovolume in SSW ($2.90\times10^8$ and $4.80\times10^7$ µm³ L⁻¹, respectively). Differences in dominant phytoplankton size fraction were apparent between stations. The 10-20 µm fraction was dominant at 2-STF (62% and 54% in the SML and SSW, respectively), and 4-SAW (85% in the SML and SSW), whereas the 20-50 µm fraction dominated at station 5-STW (42% and 49% in the SML and SSW, respectively), and in the SML at 1-STF (43%), but lowest

270 in the SML at 3-SAW (15%). The <10 µm size fraction generally accounted for the smallest biovolume (<10%), except in the SML at 6-Mixed where it was the dominant size fraction (30%). Variations were generally consistent within stations, with similar size fraction abundance in the SML and SSW, except at 1-STF which showed a lower biovolume in the >50 µm fraction and corresponding higher biovolume in the 10-20 µm fraction in the SML.





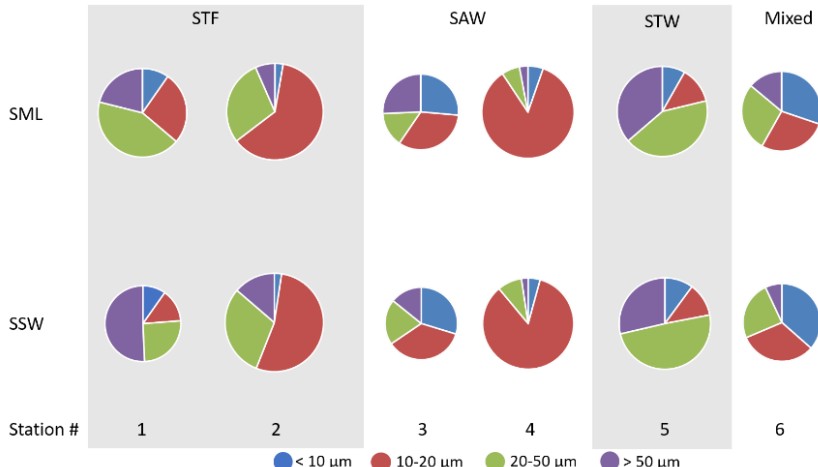

**Figure 6: Pie charts showing the variation of phytoplankton size fraction in the SML and SSW at the six stations. The size of the pie is proportional to the total summed biovolume (in µm³ L⁻¹) for phytoplankton > 5 µm, with the coloured wedges corresponding to the different size fractions. Water mass type is indicated by the label at the top of the figure and the shading.**

The composition of the phytoplankton community was also examined in terms of carbon biomass (Figure 7). Total phytoplankton biomass was highest at 1-STF, and in SSW at 2-STF and 5-STW (22 to 31 mg C m⁻³), and lowest at 3-SAW and 6-Mixed (3.9 to 8.3 mg C m⁻³). The phytoplankton groups in the SML and SSW varied with water mass, with dinoflagellates dominating at all stations, except 2-STF and 5-STW where diatoms dominated (2-STF SSW 52%; 5-STW SML 75%, SSW 61%). Dinoflagellate biomass averaged 7.4 mg C m⁻³, with a maximum at 1-STF (18 mg C m⁻³) and minimum at 5-STW (1.6 mg C m⁻³, Suppl. Info. Figure S1), whereas diatom biomass was generally lower with a maximum at 5-STW (19 mg C m⁻³) and minimum at 3-SAW (0.1 mg C m⁻³, Suppl. Info. Figure S1). The small flagellates had lower biomass (<16%), except at 3-SAW and 6-Mixed (SML 28% and 43%, respectively and SSW 9% and 37%, respectively). The dominant phytoplankton genus (>5 µm) was the dinoflagellate *Gymnodinium*, which accounted for more than 10% of total phytoplankton biomass in the SML at 1-STF, 3-SAW and 4-SAW, and 6% at 6-Mixed (Suppl. Info Figure S2).





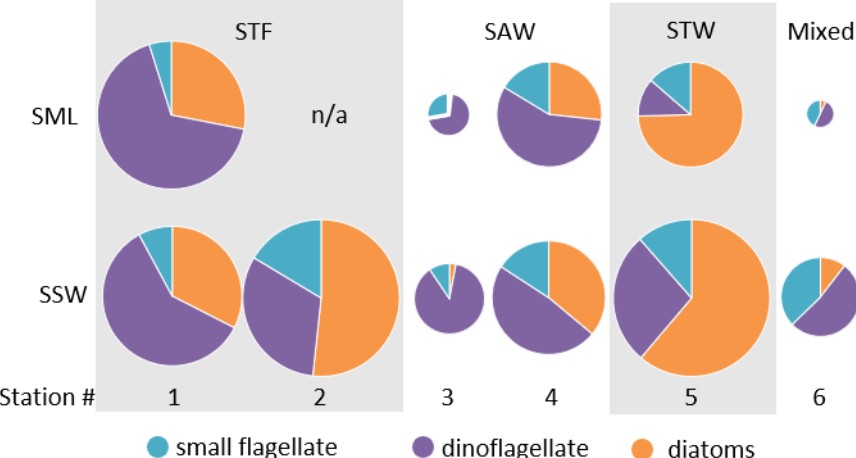

**Figure 7: Pie charts showing the variation of diatoms, dinoflagellate and small flagellates in the SML and SSW. The size of the pie is proportional to the total carbon content of phytoplankton >5 µm. Water mass type is indicated by the label at the top of the figure and also the shading. There is no data for SML 2-STF.**

There was no change in the dominant phytoplankton group between the SML and SSW at all stations. There was generally lower dinoflagellate biomass in the SML relative to the SSW (Figure 8), with an average EF of 0.75 (range: 0.19 – 1.43) with enrichment only observed at 1-STF (1.43) and 4-SAW (1.14). Diatom biomass was also lower in the SML, with an average EF of 0.62 (range: 0.31 – 1.09), with only 1-STF showing enrichment (1.09).

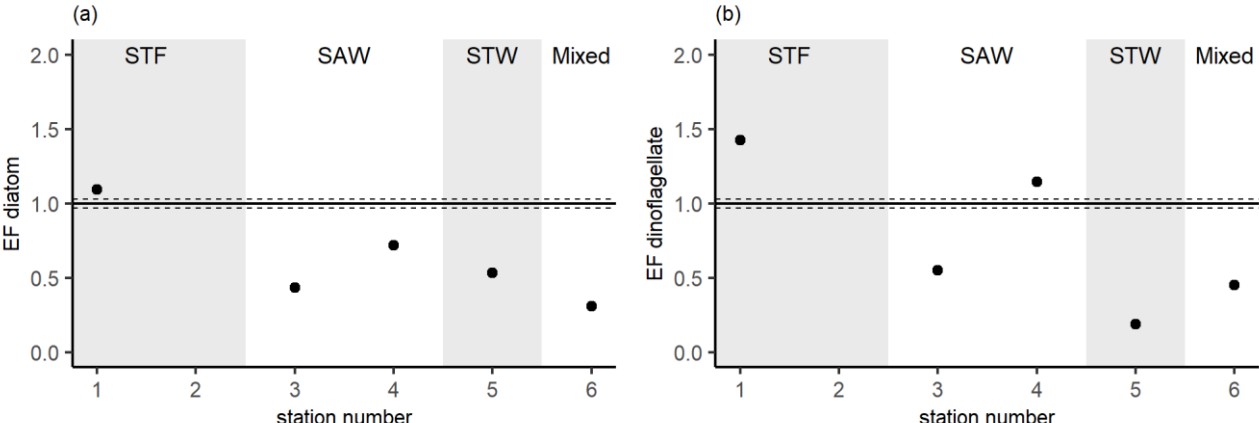

**Figure 8: EF phytoplankton carbon content for (a) diatoms and (b) dinoflagellate of >5 µm. The dashed lines represent the significance threshold determined from the reproducibility of the sipper (3%). Water mass type is indicated by the label at the top of the figure and also the shading.**

## 3.4 Correlations between variables

For all station data the Pearson test identified that DMSP concentration, bacterial abundance and diatom biomass in the SML were significantly correlated to their respective concentrations or abundance in the SSW (Table 2 and Table 3). In addition, DMSP and DMS were correlated in both the SML and SSW (Spearman's rank test in the SML, and Pearson test in the SSW;





Table 2 and Table 3). The SML DMSP concentration was also correlated with SML dinoflagellate biomass (Pearson test, Table 3). The Spearman's rank test established that chl-*a* and DMS in the SML correlated to their respective concentrations in the SSW, and DMS concentration in the SML also correlated with SML chl-*a* concentration, the 20-50 µm fraction (Spearman's rank test Table 2) and the biomass of the dinoflagellate *Gymnodinium* (rho = 0.95; p = 0.05; Spearman's rank test). The diatom biomass and 20-50 µm fraction in the SML influenced the chl-*a* concentration, as identified by correlations between these variables (Table 2). In the SSW, small flagellate biomass, 20-50 µm and >50 µm size fractions reflected the chl-*a* concentration (Table 2 and Table 3).

**Table 2: Summary of Pearson test results in the white boxes, and Spearman's rank correlation in the shaded boxes, for DMSP, DMS and all ancillary variables, with the phytoplankton sorted by both size and group. The correlations are significant when r or rho (for Pearson and Spearman's rank tests, respectively) is > 0.5 and p < 0.05, as indicated in bold. The correlations for the SML are shown in the top of the diagonal and for the SSW in the bottom of the diagonal. The size fraction biovolumes were obtained from Flowcam. N.D. stands for no data.**

| SML / SSW | DMSP | DMS | chl-*a* | <10 µm | 10-20 µm | 20-50 µm | >50 µm |
|---|---|---|---|---|---|---|---|
| DMSP | **0.95 (*<0.01*)** | **0.90 (*0.01*)** | 0.83 (*0.06*) | 0.14 (*0.78*) | 0.14 (*0.78*) | 0.59 (*0.22*) | 0.11 (*0.84*) |
| DMS | **0.84 (*0.04*)** | **0.99 (*<0.01*)** | **0.93 (*<0.01*)** | 0.24 (*0.65*) | 0.24 (*0.65*) | **0.82 (*0.04*)** | 0.22 (*0.68*) |
| chl-*a* | 0.48 (*0.33*) | 0.73 (*0.10*) | **0.94 (*0.02*)** | 0.26 (*0.66*) | 0.77 (*0.10*) | **0.89 (*0.03*)** | 0.77 (*0.10*) |
| <10 µm | -0.52 (*0.29*) | -0.08 (*0.89*) | 0.39 (*0.45*) | 0.43 (*0.39*) | N.D. | N.D. | N.D. |
| 10-20 µm | -0.52 (*0.29*) | -0.08 (*0.89*) | 0.72 (*0.11*) | N.D. | 0.43 (*0.39*) | N.D. | N.D. |
| 20-50 µm | 0.03 (*1.00*) | 0.6 (*0.24*) | **0.92 (*<0.01*)** | N.D. | N.D. | 0.83 (*0.06*) | N.D. |
| >50 µm | 0.43 (*0.42*) | 0.83 (*0.06*) | **0.91 (*0.01*)** | N.D. | N.D. | N.D. | 0.77 (*0.10*) |

**Table 3: Summary of Pearson test results in the white boxes, and Spearman's rank correlation in the shaded boxes, for DMSP, DMS and all ancillary variables, with the phytoplankton sorted by both size and group. The correlations are significant when r or rho (for Pearson and Spearman's rank tests, respectively) is > 0.5 and p < 0.05, as indicated in bold. The correlations for the SML are shown in the top of the diagonal and for the SSW in the bottom of the diagonal. Diatoms, dinoflagellates and small flagellates were obtained from optical microscopy. N.D. stands for no data.**

| SML / SSW | DMSP | DMS | chl-*a* | Diatoms | Dinoflagellates | Small flagellates |
|---|---|---|---|---|---|---|
| DMSP | **0.95 (*<0.01*)** | **0.90 (*0.01*)** | 0.83 (*0.06*) | 0.42 (*0.48*) | **0.89 (*0.04*)** | -0.08 (*0.90*) |
| DMS | **0.84 (*0.04*)** | **0.99 (*<0.01*)** | **0.93 (*<0.01*)** | 0.67 (*0.22*) | 0.67 (*0.21*) | -0.05 (*0.93*) |
| chl-*a* | 0.48 (*0.33*) | 0.73 (*0.10*) | **0.94 (*0.02*)** | 0.80 (*0.10*) | 0.40 (*0.50*) | -0.20 (*0.75*) |
| Diatoms | 0.10 (*0.84*) | 0.37 (*0.47*) | 0.72 (*0.10*) | **0.92 (*0.03*)** | N.D. | N.D. |
| Dinoflagellates | 0.75 (*0.08*) | 0.51 (*0.30*) | 0.42 (*0.40*) | N.D. | 0.84 (*0.07*) | N.D. |
| Small flagellates | -0.06 (*0.90*) | 0.44 (*0.38*) | 0.74 (*0.09*) | N.D. | N.D. | 0.44 (*0.46*) |

## 3.5 Air-sea flux

Average wind speeds over the previous 12 h ranging from 3.79 to 8.19 m s$^{-1}$ for the workboat sampling. Average DMS fluxes were 3.68 µmol m$^{-2}$d$^{-1}$ (range: 2.45 – 6.96 µmol m$^{-2}$ d$^{-1}$) for of F$_{SML}$, and 5.32 µmol m$^{-2}$ d$^{-1}$ (range: 2.49 – 11.56 µmol m$^{-2}$ d$^{-1}$), with higher F$_{SSW}$ at higher wind speeds and DMS concentrations as expected (Table 4). Air-sea flux was also calculated using DMS concentration at 5 m depth (F$_{5m}$) and compared with the SML and SSW fluxes to examine the influence of depth on





calculated flux. $F_{5m}$ resulted in an average flux of 3.87 µmol m$^{-2}$ d$^{-1}$ (range: 2.28 – 8.80 µmol m$^{-2}$ d$^{-1}$), consistent with $F_{SML}$ except at 2-STF, 4-SAW and 5-STW where $F_{SML}$ was lower. The difference in DMS air-sea flux calculated for the three

different depths was primarily due to the higher DMS concentration in the SSW.

**Table 4: DMS air-sea flux calculated using the COARE algorithm for each station.**

| Workboat station | Averaged wind speed 12 h prior to sampling (m s$^{-1}$) | [DMS] (nmol L$^{-1}$) | | | Flux (µmol m$^{-2}$ d$^{-1}$) | | |
|---|---|---|---|---|---|---|---|
| | | SML | SSW | 5 m | SML | SSW | 5 m |
| 1-STF | 3.79 | 3.08 | 4.23 | 2.95 | 2.94 | 4.04 | 2.82 |
| 2-STF | 7.50 | 3.76 | 6.24 | 4.75 | 6.96 | 11.56 | 8.80 |
| 3-SAW | 7.88 | 1.52 | 1.25 | 1.28 | 3.03 | 2.49 | 2.55 |
| 4-SAW | 7.36 | 1.69 | 4.20 | 2.20 | 3.20 | 7.96 | 4.17 |
| 5-STW | 5.36 | 1.67 | 1.78 | 1.76 | 2.45 | 2.62 | 2.59 |
| 6-Mixed | 8.19 | 1.52 | 1.41 | 1.27 | 3.51 | 3.25 | 2.28 |

## 4    Discussion

From a regional perspective, the Sea2Cloud results contrast with previous studies (Law et al., 2017; Walker et al., 2016), with
lower DMS concentrations encountered in SSW, and SML DMS enrichment at only one of the six stations. Furthermore, chl-*a* was also not enriched in the SML, contrary to that reported in other studies (Yang et al., 2009; Zhang et al., 2008; Zhang et al., 2009). Enrichment of biogeochemical variables, such as chl-*a*, DMSP and DMS, in the SML has often been observed during a phytoplankton bloom in the underlying water (Nguyen et al., 1978; Yang et al., 2005a; Zhang et al., 2009; Walker et al., 2016); yet despite the major diatom bloom of 4.3 µg L$^{-1}$ chl-*a* at 2-STF (Sellegri et al., in revision) exceeding maximum
chl-*a* concentrations recorded during the previous SOAP voyage (2.8 µg L$^{-1}$; (Lizotte et al., 2017)), it appears that bloom conditions alone are insufficient to generate chl-*a* enrichment in the SML. These contrasting regional results (Bell et al., 2015; Walker et al., 2016; Lizotte et al., 2017) suggests non-optimal conditions for DMS and chl-*a* enrichment in the SML in the current study.

SML DMSP concentration was primarily influenced by dinoflagellate biomass, as indicated by the correlation between these variables (Table 3). This is consistent with previous observations, in which DMSP enrichment in the SML was attributed to phytoplankton composition (Yang and Tsunogai, 2005; Zemmelink et al., 2006), particularly when dinoflagellates were dominant (Yang, 1999; Matrai et al., 2008; Yang et al., 2009). However, DMSP was not enriched in the SML during the SOAP voyage, despite the high dinoflagellate biomass (C. Law, pers. comm), and SML enrichment only occurred at one station in
the current study where the ratio of dinoflagellate to diatoms was lowest (5-STW, 0.2, Figure 3b). The correlation between DMSP and dinoflagellates was high in both SML and SSW in the current study, but only significant in the SML, indicating that specific factors enhance this relationship in the SML. DMSP production increases under oxidative stress (Sunda et al., 2002), and so light stress may be a co-factor that enhances DMSP production by dinoflagellates in the SML.





The complexity of DMS cycling often precludes identification of the main drivers of DMS production, and this is particularly

so in the SML where loss of DMS to the atmosphere obscures potential relationships with conservative properties such as chl-

*a* and phytoplankton group (Stefels et al., 2007; Bürgermeister et al., 1990; Townsend and Keller, 1996; Turner et al., 1988).

Indeed, only one study has previously reported a correlation between enrichment of chl-*a* and DMS in the SML (Yang and

Tsunogai, 2005). However, DMS concentration in the SML was correlated to both the chl-*a* and 20-50 µm size fraction in the

current study (Table 2). During SOAP, high DMS EF and concentrations were associated with a dinoflagellate bloom (Walker

et al., 2016) with *Gymnodinium* and *Gyrodinium* being the most abundant genera, in addition to *Ceratium* and small flagellates

(C. Law, pers. comm., Suppl. Info. Figure S3). In both SOAP and the current study, SML DMS was significantly correlated

with *Gymnodinium* (Spearman's rank test; rho = 0.95; p = 0.05 and rho = 0.76; p = 0.02, respectively, and Spearman's rank

test for both studies, rho = 0.60; p = 0.01). The relationship between DMS and dinoflagellates is consistent with dinoflagellate

being a source of DMSP, but also DMSP conversion to DMS may potentially be enhanced by other factors. For example,

copepod grazing pressure on *Gymnodinium* is reported to influence DMS concentration (Dacey and Wakeham, 1986).

Moreover, during senescence, dinoflagellates release gel-like compounds that accumulate in the SML (Jenkinson et al., 2018),

altering the physical and properties of the SML and influencing gas exchange (Wurl et al., 2016). Consequently, dinoflagellates

affect DMSP and DMS both directly and indirectly in the SML.


DMS loss is expected to be more rapid in the SML due to its proximity to the atmosphere. However, other processes such as

elevated photo-oxidation of DMS in the SML may also be part of DMS removal processes in the surface ocean (see companion

paper (Saint-Macary et al., in revision)). The DMS maximum in the SSW, relative to the SML and 5 m depth (Figure 4a) may

reflect a combination of near-surface stratification and elevated DMS ventilation at the surface. This is in contrast to the

observations of Walker et al. (2016) in the same region who reported the opposite effect, with high DMS enrichment in the

SML. The latter may have arisen from an optimal combination of factors: (i) a dinoflagellate bloom supporting elevated DMSP

and resulting DMS production (Walker et al., 2016), (ii) favourable meteorological conditions i.e. very low wind speeds (Law

et al., 2017), that limited near-surface mixing and led to (iii) near-surface stratification (Smith et al., 2018). Although near-

surface temperature measurements were not obtained during the current study, wind speeds were generally higher than during

SOAP, indicating higher mixing and reduced potential for near-surface stratification, although sole observation of DMS

enrichment occurred at the station with the highest wind speeds (see Table 4). Contrasting near-surface DMS gradients have

been reported in a stratified salt pond (Zemmelink et al., 2006) and coastal water under calm meteorological conditions

(Zemmelink et al., 2005), with respective increases and decreases in DMS concentration to the surface. The key factor

determining DMS enrichment or depletion in the SML in these studies was irradiance, which stimulated DMSP production via

the phytoplankton antioxidant response in the salt pond (Zemmelink et al., 2006), and DMS photo-oxidation in the stratified

coastal water (Zemmelink et al., 2005). Consequently, consideration of the physical controls in addition to biogeochemical

processes is required to explain DMS enrichment in the SML (assessed in a companion paper; (Saint-Macary et al., in



revision)). An additional factor influencing enrichment may be the presence of surfactant, which can act as a barrier to gas transfer (Broecker et al., 1978; Goldman et al., 1988; Pereira et al., 2016). Surfactant was enriched at half of the stations (3-
SAW, 4-SAW and 6-Mixed; T. Barthelmess, pers. comm.), one of which showed DMS enrichment in the SML, although there was no correlation between surfactant and DMS, in terms of concentration or enrichment.

The current study also highlighted variation in sampling efficiency of different methodological approaches for determining DMS enrichment in the SML. The higher DMS concentrations obtained with the gas-permeable tube relative to the glass plate
may reflect reduced exposure of the water sample to air, which is a drawback of SML techniques such as the plate, screen and rotating drum (Yang, 1999; Zhang et al., 2009; Matrai et al., 2008; Zemmelink et al., 2006). Loss to the atmosphere is generally not accounted for in other SML studies (Zemmelink et al., 2005; Yang et al., 2001). DMS is potentially lost with the gas-permeable tube, as the upper surface is exposed to the atmosphere; however, this is minimised by smearing of the SML over the tube by surface turbulence, and gas loss is accounted for by the diffusion efficiency correction (see Methods). The
performance of the plate and the screen has been shown to vary with environmental conditions (Yang et al., 2001), which may reflect that the plate samples a thinner layer than the gas-permeable tube (nominally 20-150 µm (Cunliffe et al., 2013) and 1.21 mm, respectively). However, the plate samples the organics and bacteria of the SML, which may induce in vitro reactions in the sample bottle prior to analysis that may affect DMS concentration, whereas these are excluded with the gas-permeable tube. Another advantage of the gas-permeable tube is that it eliminates exposure of the water sample to high light, as with the
plate and screen, so avoiding stress-induced responses and cell lysis. Patchiness of the SML (Frew et al., 2002; Ribas-Ribas et al., 2017) is an issue that will decrease the reproducibility of all SML sampling techniques, but the larger surface area of the gas-permeable tube may increase this variability. Yet, despite the increased effectiveness of the permeable tube technique for dissolved gases, the results indicate that DMS is not significantly enriched in the SML, in contrast to other studies that have used the plate and screen (Nguyen et al., 1978; Yang, 1999; Yang and Tsunogai, 2005; Yang et al., 2005a; Yang et al., 2001;
Zhang et al., 2009; Walker et al., 2016; Zemmelink et al., 2006). Excluding the methodological shortcomings detailed here, this anomaly may reflect differing environmental conditions between studies; however, environmental conditions are rarely reported, and only a few have considered DMS fate in the SML (Zemmelink et al., 2006; Zemmelink et al., 2005; Matrai et al., 2008; Walker et al., 2016). Consequently, it is difficult to draw conclusions as to whether previously reported DMS enrichments are artefacts, which limits the identification of the factors responsible for DMS enrichment.


DMS air-sea flux was calculated using the COARE algorithm, which was originally developed and tested based upon a representative depth of 5 m for surface waters (Huebert et al., 2004); consequently, this approach may be less appropriate for application to the SML, where conditions are not as homogenous as water at 5 m (Frew et al., 2002; Ribas-Ribas et al., 2017). Regardless, the calculated fluxes based upon three different depths were consistent, and also low relative to previous regional
measurements during the SOAP campaign, in which DMS flux reached 100 µmol m$^{-2}$ d$^{-1}$ (Bell et al., 2015; Walker et al., 2016). The large difference in flux between SOAP and the regional climatological estimate Lana et al. (2011) reflects the high



DMS concentration in the dinoflagellate bloom during SOAP; the lower DMS concentrations and emission during the current study reflect differing phytoplankton community composition and surface ocean dynamics, but also potentially different process rates (Saint-Macary et al., in revision).

## 5    Summary and conclusion

The current study presents the first application of a more accurate sampling technique for trace gases in the SML, and identified higher DMS concentrations relative to the standard SML sampling technique of the plate (Figure 4b). However, DMSP and DMS were generally not enriched in the SML, with significant DMS enrichment at only one of six stations, and low chl-*a* enrichment despite sampling of different water masses, phytoplankton biomass and community composition. However, relationships were apparent between DMSP, DMS, and both dinoflagellate and *Gymnodinium* biomass, suggesting that SML DMS production may be enhanced in the presence of dinoflagellates. These observations compliment a previous regional study indicating that an optimal combination of physical and biological conditions are required for DMS enrichment in the SML. The calculated DMS air-sea fluxes were consistent with regional estimates in the Lana et al. (2011) and Wang et al. (2020) climatology models and, indicate that DMSP and DMS cycling in the SML do not significantly influence regional air-sea DMS flux. These results raise questions about the significance of DMS enrichment in the SML and also how this can be maintained at the ocean interface where loss to the air dominates, and so emphasises the need for DMS process studies in the SML (Saint-Macary et al., in revision).

**Acknowledgment.** We would like to thank Antonia Cristi and Wayne Dillon for their help during the Sea2Cloud campaign, and Karen Thompson for Flow Cytometry analysis. This research was supported by NIWA SSIF funding to the Ocean-Climate Interactions Programme. We would also like to thank the support and expertise of the Officers and Crew of the R/V Tangaroa.

**Author contribution.** Alexia D Saint-Macary and Theresa Barthelmeß developed the sampling strategy. Alexia D Saint-Macary wrote the manuscript, analysed DMSP and DMS. Andrew Marriner contributed to DMSP and DMS analysis. Theresa Barthelmeß prepared and analysed samples for the Flow Cytometry and organic variables. Stacy Deppeler analysed samples on the Flowcam and processed results, and Karl Safi identified the species by optical microscopy. Nutrient and chl-*a* data were collected by Antonia Cristi, Karl Safi and Stacy Deppeler. Rafael Costa Santana and Mike Harvey calculated the DMS air-sea flux. Alexia Saint-Macary, interpreted the results, with guidance from Cliff Law. There are no competing interests.

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
