# Peer review of "DMS cycling in the Sea Surface Microlayer in the South West Pacific: 1. Enrichment potential determined using a novel sampler"

_EGUsphere, 2022_

## Author Comment (AC1)

Egusphere-2022-499: **DMS cycling in the Sea Surface Microlayer in the South West Pacific: 1. Enrichment potential determined using a novel sampler**

Answers to reviewer 1

The authors would like to thank anonymous reviewer 1 for the very positive review provided. The answers to the comments are provided in blue below.

Comment 1: Sections '2.6 DMS air-sea flux calculation' and '3.5 Air-sea flux': DMS concentrations in the SML were determined using the gas-permeable tube, the plate and the sipper techniques. Thus, DMS air-sea flux estimates can be obtained using DMS concentrations from three different sampling methods. Please point out detailed DMS concentrations which were used in DMS air-sea flux calculation. DMS measurements with the gas-permeable tube, or with the plate, or with the sipper?

Answer: L202-203 regarding the sampler used for SML DMS air-sea flux "$F_{SML}$ corresponds to DMS air-sea flux calculated using SML DMS concentration obtained with the gas-permeable tube", and in Table 4 caption.

Comment 2: Section '3.4 Correlations between variables': I thought DMS measurements with the plate, or with the sipper, were used to the Pearson test. Please point out it clearly. Moreover, a novel gas-permeable tube technique approach gave accurate measurements of DMS concentrations in the SML. Why did not choose DMS measurements with the gas-permeable in the SML to analyze correlations between variables?

Answer: In the Table 4 caption, a sentence was added regarding the sampler used for SML DMS concentration "SML DMS was sampled with the gas-permeable tube", and in the text L302-303 "The SML DMS concentration presented in this section was obtained from the gas-permeable tube and was not normally distributed."

---

## Author Comment (AC2)

Egusphere-2022-499: **DMS cycling in the Sea Surface Microlayer in the South West Pacific: 1. Enrichment potential determined using a novel sampler**

Answers to reviewer 2

The authors would like to thank anonymous reviewer 2 for the comments and suggestions provided. Below in blue are the answers to the comments.

**Abstract**

In the abstract, it would be good to mention:

- that the SSW was sampled at a depth of 0.5 m **and 5m**
- that "enrichment" refers to "enrichment **of the SML**" relative to the SSW at a depth of 0.5 m
- outcomes of the method comparison between the gas-permeable tube and glass plate techniques

Answer: the chl-*a* concentration at 5 m was removed from the manuscript. DMS concentration is only presented at 0.5 m, and so there is no need to mention sampling at 5 m.

"enrichments in the SML relative to the SSW were only …" was clarified L14.

Outcomes of the method comparison between the gas-permeable tube and the plate is addressed in L16 "DMS in the SML was determined using a novel gas-permeable tube technique which measured consistently higher concentrations than with the traditional glass plate technique…".

**Introduction**

**Line 25-27:** I suggest being careful with the wording here: DMS is *mainly* derived from DMSP but not exclusively (e.g. DMSO is another precursor of DMS).

Answer: modification done.

Also, DMSP is not only exclusively produced by phytoplankton but by other marine algae and higher plants (Stefels 2000), coral (Raina et al 2013) and bacteria (Curson et al., 2017). Unless the authors want to specifically describe the main source of DMSP in the SML? In which case, this should be mentioned and at the very least bacteria as a source of DMSP should also be accounted for.

Answer: the sentence was modified to "…DMSP primarily produced by phytoplankton…".

Not all of the DMS produced is ventilated to the atmosphere. Only about ~ 10% of it (Malin et al., 1992).

Answer: modification done L28 "About 4 to 16% of DMS is ventilated to the atmosphere (Galí, et al., 2015)…".

**Line 30:** "…and consequently *decreasing/inhibiting* phytoplankton growth…."?

Answer: modification done "and consequently decreasing phytoplankton growth...".

**Line 33:** "…elucidate *potential* feedbacks *of marine gas/DMS emissions on climate*."?

Answer: modification done "it continues to be investigated to elucidate potential feedbacks of DMS emissions on climate.".

**Line 42:** "…affect the *DMS* flux."?

Answer: modification done.

**Line 66-68:** please describe what these 2 approaches are so the reader doesn't have to read Walker et al 2016 to source the information.

Answer: "with two independent approaches (direct SML concentration measurement and indirect estimation from eddy covariance) indicating that DMS enrichment in the SML influenced air-sea flux {Walker, 2016 #237}."

**Line 70:** why the maintenance?

Answer: the sentence was re-written as "The SOAP results also raised questions regarding how DMS enrichment is maintained in the SML, on DMS emissions and its influence on DMS emissions."

**Line 71-74:** here it would be helpful to mention that Walker et al. 2016 (SOAP Voyage) used both the plate and Garret screen methods in their study, and showed that the screen method led to an **overestimation** of DMS concentration. (On line 73 the authors are saying that the Garret screen may lead to an **underestimation** of DMS concentration. Did they mean **overestimation**?).

Answer: The studies cited L77 acknowledge that the plate and the screen underestimate DMS concentrations. Walker et al., (2016) found that during SOAP organic matter was preconcentrating in the mesh which led to overestimation of DMS. Another sentence was added L78-79 for more clarity. "However, Walker et al., (2016) used the plate and the Garret screen

and found that the screen was overestimating DMS due to preconcentration of organic material in the mesh.".

**Line 74-76:** This sentence is confusing. Did you mean: "To address this, a novel SML sampling technique using gas-permeable tube to minimize DMS loss was deployed ***during the Sea2Cloud voyage***, and results compared to those obtained with the glass plate method **used during the SOAP Voyage**."?

Answer: during Sea2Cloud, the gas-permeable tube method was used to sample the SML, but as it is a novel method, the plate was also used as a comparison, as shown in Figure 4b. The end of the sentence was removed for clarity "To address this, a novel SML sampling technique using gas-permeable tube to minimize DMS loss was deployed during the Sea2Cloud voyage, and results compared to those obtained with the glass plate method used during the SOAP Voyage.".

**Line 78:** "…*estimate* EFs…"? Are you generating or estimating EFs through this study?

Answer: estimate, modification done.

**Line 79**: (see companion paper, Saint-Macary et al., ***in revision***?)

Answer: The manuscript number was added, and the companion paper is now cited as "Saint-Macary et al., eguspehere-2022-504". Hopefully, the two manuscripts will be published at the same time so the proper reference will be cited.

**Method**

**Line88-89:** Do you mean that SSW was collected with a Niskin bottle or a rosette? CTD only means "conductivity – temperature – depth". Also, it would be good to mention the depth (0.5 m? as mentioned in the abstract? Or 5 m as mentioned in the "CTD sampling" section)

Answer: This sentence was removed "SSW measurements were complemented by surface water collection using a CTD" as it introduced confusion and is not necessary in this section. The sampling section is described below. But to clarify, the SML and SSW was sampled from the workboat using the gas-permeable tube and a Teflon tube, respectively, for DMS. The CTD was deployed after the workboat to sample at 5 m depth.

**Table 1:** minor comment: it would look better to have the sampling time and average windspeed on one line instead of 2? (maybe by playing with font size)

Answer: The font size of the sampling time was decreased, and the average wind speed was put on the same line instead of 2.

**Line 99-100:** On the 26th of March sampling time was 09:50am - 11:38am, so later than 11:00. Maybe you could say in 0800 and 1200? What was the approximate distance of the workboat from RV Tangaroa?

Answer: The distance was added, and the sampling time range modified. "The SML and SSW were sampled from a workboat at 0.5 to 1 nautical mile away from the R/V *Tangaroa* between 0800 and 1200 (all times NZDT)…".

**Line 100:** There were only 6 sampling days out of a 13-day voyage, so I would write "on each *sampling* day". Why sampling at 5-STW occurred at a different time of the day? It's not clear if it was because of wind speed. Please specify.

Answer: "each sampling day was removed from L100, and the specification re 5-STW sampling time was added to L105 "Station 5-STW was sampled in the afternoon due to high wind speed in the morning ($> 10$ m s$^{-1}$)."

**Line 121:** "…using Eq. (1)"
Answer: modification done (now L129).

**Line 123:** "…where [DMS]MQ and [DMS]tank are the DMS concentrations in nmol L-1 measured in the MQ and the calibration tank *at t+10min*, respectively." Is that correct? Was there any difference between DMS concentration in the calibration tank at t0 and t10? It would be important to mention this result, even if it is not shown.

Answer: DMS concentration in the calibration tank was slightly different between $t_0$ and $t_{+10 \, min}$, therefore the average of the two concentrations was used to calculate the diffusion efficiency. It is now clarified L131-132 "where [DMS]$_{MQ}$ is the DMS concentration measured in the MQ at $t_{+10 \, min}$, and [DMS]$_{tank}$ is the averaged DMS concentration between $t_0$ and $t_{+10 \, min}$ measured in the calibration tank."

**Line 126:** "…in Saint-Macary (2022)."
Answer: modification done.

**Line 127:** "…using Eq. (2)"

Answer: modification done.

**Line 138:** "…following Eq. (3)"
Answer: modification done.

**Line 139**: I am questioning the validity of this equation. Reproducibility is usually measured as the standard deviation of the difference between multiple measurements. Please provide source for equation 3.

Also, the term "reproducibility" is commonly used when comparing the difference in measurements from different laboratories using the same technique whereas "repeatability" is more commonly used to describe the difference in measurements between different techniques within the same laboratory.

Also, it would be good here to give the level of replication, which I believe is 5-6 based on paragraph 211-221.

Answer: The reviewer is correct that "repeatability" is a more correct term for our estimate of method precision, which describes the relative percentage difference between a pair of repeated measurements, and so we have corrected this throughout.

L145 now say "the repeatability, estimated as the relative percentage difference between a pair of repeated measurements, for the gas-permeable tube……"

**Line 140-145:** I think this section and the "CTD sampling" should go under "2.3. Sampling of the SSW", so then it is clear that the SSW was sampled with the Teflon tube at 0.5m and with the CTD Rosette at 5m depth.
Answer: modification done.

**Line 147:** since the SML sampling didn't always happen at the same time (especially at 5-STW and 6-Mix), it would make more sense to say that "…CTD sampling occurred in **between XXh and XXh following SML sampling**."
Answer: "The CTD was launched between 1000 and 1215 following SML sampling, except at 5-STW when the CTD was deployed before the SML sampling at 0700."

**Line 193:** Please provide source for equation 4
Answer: the reference "Liss and Merlivat, 1986" was added L201.

**Results**

**Line 212-214:** It seems like the lowest reproducibility for DMSP was obtained with the sipper and not the plate. I thus don't understand this sentence.

Answer: The sentence was re-written for more clarity: "The median repeatability of DMSP measurement at the six workboat stations was 7% (interquartile range: 4 – 8%) for the plate, whereas the sipper had a higher median repeatability of 3% (interquartile range: 1 – 7% n = 5)".

**Caption, Fig.3:** "(a) DMSP concentrations, sampled in the SML **and in the SSW by** the sipper"

Answer: for DMSP measurements, the SML was sampled with the sipper in Fig 3a, and the SSW with the Teflon tube. To clarify, the sipper was added in the caption as Fig 3b shows enrichment factors from the sipper and the plate.

I would remove "depth" and "method" at the bottom of each graph as it is clear what is being compared here.

Answer: "depth" and "method" were removed from the figures. For coherency between the companion papers, the shaded area corresponding to the different water types were replaced by vertical dashed lines, as suggested by another reviewer.

**Lines 243-245:** Or could it be that the gas-permeable tube leads to an overestimation of DMS concentrations? This is consistent with the Garret screen/plate comparison in Walker et al. 2016 (lower concentrations measured by the plate)

Answer: In our comparison the gas-permeable tube gives a higher SML DMS concentration than the plate. Reports from the literature indicate that the plate is underestimating DMS concentration as there is loss to the atmosphere. There is no reason or evidence to indicate that the gas-permeable tube overestimates concentration, and the DMS EF from the gas-permeable tube is not significantly higher than that found in other studies with the plate or the screen. Therefore, we are confident that the gas-permeable tube is not overestimating DMS concentration in the SML. We note also that the Garret screen-plate result in Walker et al., (2016) cannot provide any insight into the effectiveness of the gas-permeable tube in this study.

**Caption, Fig.4:** "(a) DMS concentrations in the SML, SSW, **and at** 5 m **depth**…"
Answer: modification done.

Same here, I would remove "depth" and "method" at the bottom of each graph so it doesn't get mixed up with the legend.

Answer: "depth" and "method" were removed from the figures. For coherency between the companion papers, the shaded area corresponding to the different water types were replaced by vertical dashed lines, as suggested by another reviewer.

**Line 251-260:** why is it that EF for chl-a is relative to 5m and 0.5 SSW whereas DMS and DMSP EF are only relative to 0.5 m? Same for the phytoplankton community: it seems like only the populations of the SML and SSW at 0.5m depth are compared. Is there a point showing the data at 5m depth if it's only used for one variable?

Answer: chl-*a* EF relative to 5 m depth has been removed as this was not discussed. Chl-*a* concentration at 5 m was also removed from Fig 5a.

**Line 256:** Suggest adding the value for the "very high enrichment at 2-STF (XX)" (2.88, is that right?)

Answer: the sentence was deleted as EF chl-*a* relative to 5 m was removed from the manuscript.

**Caption, Fig.5:** Same comment as for Fig. 3 and 4: "5 m depth" and removing "depth" and "EF relative to" labels at the bottom.

Answer: "depth" and "EF relative to" were removed from the figures. For coherency between the companion papers, the shaded area corresponding to the different water types were replaced by vertical dashed lines, as suggested by another reviewer.

**Fig. 7:** what's the reason for SML 2-STF data being missing? (please, provide brief explanation, either here or in the method section)

Answer: "sample was not obtained" was added in the Figure legend.

**Line 307:** "…and the biomass of the dinoflagellate *Gymnodinium* (rho = 0.95; p = 0.05; Spearman's rank test**; Supplementary material**)"

Answer: modification done.

**Line 307-309:** this is interpretation, and it should be left to the discussion. Also, Table 2 provides information on fractions and not groups, so why is the diatom biomass mentioned here?

Answer: the sentence was removed.

**Line 309-310:** I didn't understand this sentence? Is that based on the correlation between SSW chla and the 20-50 μm fraction in Table 2? I cannot see the correlation with the >50 μm size.

Answer: the sentence was re-written and corrected as small flagellates do not correlate with chl-*a* in the SSW, and a new sentence was added: "In the SSW, 20-50 μm and >50 μm size fractions correlated with the chl-*a* concentration (**Error! Reference source not found.**)".

**Table 2. caption**: here phytoplankton is sorted by size not group.

Answer: not relevant anymore as I modified Table 2 and 3.

**Table 3. caption**: here phytoplankton is sorted by group not size.

Answer: not relevant anymore as I modified Table 2 and 3.

**Table 2 & 3**: just to be clear, are these correlations within or between water masses? (e.g. does **0.93 (<0.01) in Table 2,** Row 2, column 4 represents a correlation between DMS and chl a in the SML or between SML's chla and SSW's DMS?)

If it's the latter, it's a bit of a stretch to correlate e.g. chla in the SSW with the microbial size fraction in the SML. If it's the former, I suggest making 2 tables (2 and 3) for SML and SSW with all the ancillary variables (both microbial fractions and groups). Otherwise, this part in Table 3 is a repeat of table 2:

Answer: these correlations using data from all stations for just the SML in Table 2 and just the SSW in Table 3. So 0.93 (<0.01) in Table 2, row 2, column 4 represents a correlation between DMS SML and chl-*a* SML. The Table 2 and 3 legends were modified as suggested to highlight this.

**Line 324:** "ranged"? (not ranging). And why "over the previous 12h."? With respective to what time?

Answer: modification done. The air-sea flux was calculated over the 12 h prior to sampling the SML as the SML structure and near-surface mixing would be influenced by winds over a longer preceding period than instantaneous winds. This clarification was added to the manuscript L359.

**Line 325:** "…were 3.68 μmol m-2d-1 (range: 2.45 – 6.96 μmol m-2 d-1) **for the SML** and 5.32 μmol m-2 d-1 (range: 2.49 – 11.56 μmol m-2 d-1) **for the SSW"?**

Answer: correct, it is clarified after the flux range with $F_{SML}$ and $F_{SSW}$ which refers to DMS air-sea flux calculated with SML DMS concentration and SSW DMS concentration, respectively, as stated L221-222 in Method.

**Line 326:** this is also true for other water masses. I would rephrase this argument: "…with generally higher DMS fluxes recorded at higher wind speeds combined with higher DMS concentrations."

Answer: modification done.

**Line 328:** "F5m resulted in an average flux of 3.87 μmol m-2 d-1 (range: 2.28 – 8.80 μmol m-2 d-1), **which was** consistent with **the average** FSML."

Answer: modification done.

**Line 329:** why do you only concentrate on data points where FSML was lower than F5m? What about times when FSML was higher? And does it really matter since the average flux for SML and 5m were similar?

If so, you could just say "although FSML and F5m exhibited differences across workboat stations, average F5m (average 3.87 μmol m-2 d-1, range: 2.28 – 8.80 μmol m-2 d-1) **was** consistent with **the average** FSML."

Answer: the sentence was modified as suggested. $F_{SML}$ was higher than $F_{5m}$ at 3 stations, and lower at the other 3 stations, so it was not making sense to only focus on the lower fluxes as the averages are similar. The sentence now reads: "Although $F_{SML}$ and $F_{5m}$ exhibited differences across workboat stations, the average $F_{5m}$ 3.87 μmol m$^{-2}$ d$^{-1}$ (range: 2.28 – 8.80 μmol m$^{-2}$ d$^{-1}$) was consistent with the average $F_{SML}$".

**Discussion**

**Line 335-336:** At station 2 Chl-a was enriched in the SML relative to 5m. This is confusing to then read this sentence in the discussion. It might be better not to present the chla EF relative to 5m depth. See previous comment.

Answer: chl-*a* concentration at 5 m has been removed to avoid confusion and because it was not discussed in the paper.

**Line 339-341:** I think it would be better rephrased as "…however it appears that the major diatom bloom of 4.3 μg L-1 chl-*a* at 2-STF (Sellegri et al., in revision), which exceeded the

maximum chl-*a* concentrations recorded during the previous SOAP voyage (2.8 µg L-1; (Lizotte et al., 2017)), was insufficient to generate chl-*a*, DMS or DMSP enrichment in the SML."

Answer: the sentence was modified as suggested.

**Line 348-350:** so what does it say about the correlation between DMSP and dinoflagellates abundance? Was it a positive or negative correlation? In fact, I think it is very important to mention whether correlations are negative or positive in your results section.

Answer: L383 now says "SML DMSP concentration was primarily influenced by dinoflagellate biomass, as indicated by the positive correlation between these variables". A sentence was added at the end of the correlation section, L319, saying that all the correlations were positive.

**Line 350:** "the lowest"

Answer: modification done.

**Line 353:** it would be good to read more suggestions of what these specific factors may be. Light availability? nutrients? Or is it that DMSP concentration in the SSW is dictated by the presence of other microbes?

Answer:  the possibility of light being a factor enhancing the relationship between DMSP and dinoflagellate in the SML is discussed in the following sentence. This seems a likely factor in the SML but we do not have data to support any data to support other factors and so do not wish to speculate.

**Line 363-364:** what does "Spearman's rank test for both studies, rho = 0.60; p = 0.01" correspond to? Is it by pulling the data together? Maybe this value is not necessary?

Answer: yes, it is pulling the data together. The coefficients were removed.

**Line 363:** I would remove "pressure"

Answer: modification done.

**Line 368:** Physical and…?

Answer: only physical, the "and" was removed.

**Line 379:** So which temperature did you use for your flux calculation in Equation (5) then?

Answer: seawater temperature from 5 m depth and measured with the underway system was used for the flux calculation and averaged over the previous 12 h of sampling (as for wind speed). It was clarified L207 "Schmidt number (Sc) calculated using local temperature (T) in °C (Saltzman et al., 1993) measured from the underway system at 5 m depth, following Eq. (5).". The average temperature was added in Table 4.

**Line 389:** What was the surfactant?

Answer: added L397-398 "Surfactant, measured in mg $L^{-1}$ TX-100 equivalents (Sigma Aldrich, TritonX 100), was enriched…".

**Line 391:** data not shown?

Answer: data not shown as it will be part of another paper.

**Line 395:** "…SML **sampling** techniques…"

Do you mean that the gas-permeable tube allows for a reduced exposure of the water sample to air? Or is it the other way around? On line 398-399 you say that "DMS is potentially lost with the gas permeable tube, as the upper surface is exposed to the atmosphere"…so in that case why do you say in line 394 that higher DMS concentrations are obtained with the gas permeable tube relative to with the plate?

Answer: yes, the gas-permeable tube reduces exposure of the water sample to the air relative to the plate, and so returns a higher DMS concentration than the plate. The text has been adjusted slightly to clarify this: "The higher DMS concentrations obtained with the gas-permeable tube relative to the glass plate may reflect that the water sample in the tubing is less exposed to air during the sampling procedure than with techniques such as the plate, screen and rotating drum (Yang, 1999; Zhang et al., 2009; Matrai et al., 2008; Zemmelink et al., 2006). Loss to the atmosphere is generally not accounted for in other SML studies (Zemmelink et al., 2005; Yang et al., 2001). Although DMS is potentially lost with the gas-permeable tube, as the upper surface is exposed to the atmosphere….".

**Line 401:** I don't understand how the variation due to environmental conditions reflects that the plate samples a thinner layer than the gas-permeable tube. You might have to develop this argument so it makes sense.

Answer: the sentence was re-written for more clarity as "when sampled with the plate and the screen the EF DMS was shown to be affected by environmental conditions and sampling thickness (Yang et al., 2001). As the plate samples a thinner layer than the gas-permeable tube (nominally 20-150 µm (Cunliffe et al., 2013) and 1.21 mm, respectively), this may result in a lower DMS concentration, depending on the SSW concentration."

**Line 407:** if anything, I would have thought that the gas-permeable tube would have decreased this variability as it allows to sample a larger surface area.

Answer: yes, the variability is decreased when the gas-permeable tube is used as it allows to sample a larger surface area. The sentence was corrected.

**Line 410-414:** But I thought that you had also used the plate for this study. Can you then compare your plate data with other studies that have also used the plate sampling technique?

Answer: the plate data was only used in our study for comparison with the gas-permeable tube with the result that the plate underestimates SML DMS concentration. Consequently, we do not feel it is of value to compare the plate data with other reported plate data.

**Line 423:** It is hard to judge without being able to read Saint-Macary et al but I would tune this argument down: "most likely reflect" or "may reflect"?

Answer: "may reflect" was written. The manuscript number of the companion paper was added, and hopefully they will be published at the same time.

**Line 426:** Is it really more accurate? I thought that both the plate and gas-permeable tube showed good reproducibility and agreement with actual DMS/DMSP concentrations measured by GC analysis during method testing?

Answer: this sentence has been clarified: "The current study presents the first application of a more robust sampling technique for trace gases in the SML…".

**Line 428:** wasn't there DMS enrichment at station 6 as well?

Answer: yes, but enrichment was not significant as it is within the repeatability threshold. In the result section, only one significant enrichment is stated, see L251 "The gas-permeable tube showed no DMS enrichment at 5 of the 6 stations, with only 3-SAW showing significant SML enrichment".

Also, there was chla enrichment at station 1, 2 (relative to 5m depth only), 3 and 4, right (relative to SSW only)?

Answer: chl-*a* enrichments relative to 5 m are now removed from the manuscript. Chl-*a* relative to SSW is significantly enriched at station 3-SAW only, as 4-SAW is within the repeatability threshold, as stated L265 "Average EF chl-*a* was 1.06 (range: 0.50 – 2.90), with significant enrichment in the SML only at 3-SAW (EF = 1.06), whereas the EF at 1-STF and 4-SAW were within the error for the sipper (**Error! Reference source not found.**b).".

**Line 430:** *Gymnodinium* is a dinoflagellate.

Answer: the sentence was modified "… dinoflagellate biomass and the genus *Gymnodinium* biomass…".

**Line 431:** DMS **and DMSP** production

Answer: modification done.

**Line 431:** by "regional", do you mean in the same region?

Answer: yes, the same region. The sentence was modified accordingly.

**Line 431:** I guess that the authors meant "complement"?

Answer: modification done.

**Line 434:** Why not since you say on line 330 that "The difference in DMS air-sea flux calculated for the three different depths was primarily due to the higher DMS concentration in the SSW" and that "higher FSSW were found at higher wind speeds and DMS concentrations" (line 326)?

Answer: the initial hypothesis was "is the SML affecting DMS air-sea flux?" and the results show the answer is no as the respective range of $F_{SML}$ is the same as $F_{5m}$, and neither differ from the climatological estimates of Lana et al., (2011) and Wang et al., (2020) climatologies. Despite $F_{SSW}$ being higher than $F_{SML}$ and $F_{5m}$ at some stations, there was no enrichment in the SML and so $F_{SML}$ was not affected by the SML.

**General Conclusion and summary comment:** maybe it would be important to talk about DMSP enrichment as well here. DMSP is not volatile (so there is no loss to the atmosphere), however there was no DMSP enrichment either. How do you interpret that?

Answer: added L439 "However, DMSP and DMS were generally not enriched in the SML, with significant enrichment of both species observed at only one of six stations…". the relationship between DMSP and dinoflagellate is stated in the following sentence.

**Supplementary material:** my suggestion is the same as for other figures: I would remove "depth" and "species" from the legend as it is clear what is being compared.

Answer: "depth" and "method" were removed from the figures. For coherency between the companion papers, the shaded area corresponding to the different water types were replaced by vertical dashed lines, as suggested by another reviewer. The abbreviations such as SML, SSW, STF, SAW and STW were also defined as it was not done in the Supplementary information.

---

## Referee Report (RR1)

**Referee Report: Reviewer 2**

**Egusphere-2022-499:** DMS cycling in the Sea Surface Microlayer in the South West Pacific: 1. Enrichment potential determined using a novel sampler

I thank the Authors for their response to my comments and suggestions. I know there were many as I took the time to review this work to the best of my knowledge and with the care that it deserves. The authors have addressed most of my comments carefully, however, it was frustrating to see that the line numbers in the authors' response did not match the line numbers in the revised manuscript nor in the tracked changes document, which make the revision process tedious.

I thank the authors for taking my comment on repeatability versus reproducibility into account. However, the following comment was only partly answered by the Authors. As this was one of my main concerns in regards to accepting this manuscript for publication, I would really appreciate the authors to reply fully to this comment and/or amend this point in the manuscript:

**Reviewer's comment**: Line 139 (**145 of revised manuscript**): I am questioning the validity of this equation. **Reproducibility is usually measured as the standard deviation of the difference between multiple measurements. Please provide source for equation 3.**

Also, the term "reproducibility" is commonly used when comparing the difference in

measurements from different laboratories using the same technique whereas "repeatability" is

more commonly used to describe the difference in measurements between different techniques

within the same laboratory.

**Also, it would be good here to give the level of replication, which I believe is 5-6 based on**

**paragraph 211-221 (220-230 of revised manuscript).**

**Author's answer**: The reviewer is correct that "repeatability" is a more correct term for our estimate of method precision, which describes the relative percentage difference between a pair of repeated

measurements, and so we have corrected this throughout.

L145 now say "the repeatability, estimated as the relative percentage difference between a pair

of repeated measurements, for the gas-permeable tube……"

Although in both cases, the smaller the number, the higher the repeatability, and the higher the reliability of the results, this is a non-conventional way of reporting repeatability. Therefore, this should either be amended or a valid source should be provided for equation 3.

---

## Referee Report (RR2)

Egusphere-2022-499: DMS cycling in the Sea Surface Microlayer in the South West Pacific: 1. Enrichment potential determined using a novel sampler

I am glad that we agree on the definition of repeatability and that this term has been amended throughout the manuscript.

However, I believe that the authors still haven't provided a source for the equation that they are using on line 146 of the revised manuscript.

This equation could be an alternative way of calculating repeatability, which I am not aware of, but if so, I would like to see a paper that uses this equation.

Here is a link to many publications in which repeatability is calculated and reported.

https://www.sciencedirect.com/topics/agricultural-and-biological-sciences/repeatability#:~:text=Repeatability%20(or%20test%E2%80%93retest%20reliability,a%20short%20period%20of%20time.

---

## Author Response (AR2)

Egusphere-2022-499: **DMS cycling in the Sea Surface Microlayer in the South West Pacific: 1. Enrichment potential determined using a novel sampler**

Answers to reviewer 2

The authors would like to thank reviewer 2 for the time and effort put into the manuscript review. We apologies for the incorrect line numbers in the authors' response.

Reviewer's comment: Line 139 (145 of revised manuscript): I am questioning the validity of this equation. Reproducibility is usually measured as the standard deviation of the difference between multiple measurements. Please provide source for equation 3.
Also, the term "reproducibility" is commonly used when comparing the difference in measurements from different laboratories using the same technique whereas "repeatability" is more commonly used to describe the difference in measurements between different techniques within the same laboratory.
Also, it would be good here to give the level of replication, which I believe is 5-6 based on paragraph 211-221 (220-230 of revised manuscript).

Initial authors' answer: The reviewer is correct that "repeatability" is a more correct term for our estimate of method precision, which describes the relative percentage difference between a pair of repeated measurements, and so we have corrected this throughout.
L145 now say "the repeatability, estimated as the relative percentage difference between a pair of repeated measurements, for the gas-permeable tube……"

Complementary authors' answer: We describe the repeat application of the same technique by the same operator with minor temporal difference in sampling time; therefore, this is not repeat measurement of a single discrete water sample under identical conditions, and so does not meet the criteria for reproducibility.
Furthermore, this differs from the reproducibility criteria (identified by the reviewer) in that there were only two samples, and it is not repeated by different laboratories/operators.
Instead, we use repeatability as described in Equation 3, which is more appropriate based on the definition of the Association for Computing Machinery, 2016, "*Same team, same experimental setup*: *The measurement can be obtained with stated precision by the same team using the same measurement procedure, the same measuring system, under the same operating conditions, in the same location on multiple trials.*" The reference was added L143 of the revised manuscript with tracked changes on.

Association for Computing Machinery (2016). *Artifact Review and Badging*. Available online at https://www.acm.org/publications/policies/artifact-review-badging (accessed October 21, 2022)